# BOSH: An Efficient Meta Algorithm for Decision-based Attacks

## Abstract

Adversarial example generation becomes a viable method for evaluating the robustness of a machine learning model. In this paper, we consider hard-label black-box attacks (a.k.a. decision-based attacks), which is a challenging setting that generates adversarial examples based on only a series of black-box hard-label queries. This type of attacks can be used to attack discrete and complex models, such as Gradient Boosting Decision Tree (GBDT) and detection-based defense models. Existing decision-based attacks based on iterative local updates often get stuck in a local minimum and fail to generate the optimal adversarial example with the smallest perturbation. To remedy this issue, we propose an efficient meta algorithm called BOSH-attack, which tremendously improves existing algorithms through Bayesian Optimization (BO) and Successive Halving (SH). In particular, instead of traversing a single solution path when searching an adversarial example, we maintain a pool of solution paths to explore important regions. We show empirically that the proposed algorithm converges to a better solution than existing approaches, while the query count is smaller than applying multiple random initializations by a factor of 10.

## 1 Introduction

It has been shown that machine learning models, including deep neural networks, are vulnerable to adversarial examples (Goodfellow et al., 2014; Szegedy et al., 2013; Chen et al., 2017a). Therefore, evaluating the robustness of a given model becomes crucial for security sensitive applications. In order to evaluate the robustness of deep neural networks, researchers have developed "attack algorithms" to generate adversarial examples that can mislead a given neural network while being as close as possible to the original example (Goodfellow et al., 2014; Moosavi-Dezfooli et al., 2016; Carlini & Wagner, 2017b; Chen et al., 2017b). Most of these attack methods are based on maximizing a loss function with a gradient-based optimizer, where the gradient is either computed by back-propagation (in the white-box setting) or finite-difference estimation (in the soft-label black-box setting). Although these methods work well on standard neural networks, when it comes to complex or even discontinuous models, such as decision trees and detection-based defense models, they cannot be directly applied because the gradient is not available.

Hard-label black-box attacks, also known as decision-based attacks, consider the most difficult but realistic setting where the attacker has no information about the model structure and parameters, and the only valid operation is to query the model to get the corresponding decision-based (hard-label) output (Brendel et al., 2017). This type of attacks can be used as a "universal" way to evaluate robustness of any given models, no matter continuous or discrete. For instance, Cheng et al. (2018); Chen et al. (2019a) have applied decision-based attacks for evaluating robustness of Gradient Boosting Decision Trees (GBDT) and random forest. Current decision-based attacks, including Brendel et al. (2017); Cheng et al. (2018); Chen et al. (2019b); Cheng et al. (2019), are based on **iterative local updates** – starting from an initial point on the decision surface, they iteratively move the points along the surface until reaching a local minimum (in terms of distance to the original example). The update is often based on gradient estimation or some other heuristics. However, the local update nature makes these methods sensitive to the starting point. As we demonstrate in Figure 1(a), the perturbation of converged adversarial examples for a neural network are quite different for different initialization configurations, and this phenomenon becomes more severe when it comes to discrete models such as GBDTs (see Figure 1(b)). This makes decision-based attacks converge to a sub-

optimal perturbation. As a result, the solution cannot really reflect the robustness of the targeted model.

To overcome these difficulties and make decision-based attacks better reflect the robustness of models, we propose a meta algorithm called BOSH-attack that consistently boosts the solution quality of existing iterative local update based attacks. Our main idea is to combine Bayesian optimization, which finds solution closer to global optimum but suffers from high computation cost, with iterative local updates, which converges fast but often get stuck in local minimum. Specifically, given a decision based attack $\mathcal{A}$, our algorithm maintains a pool of solutions and at each iteration we run $\mathcal{A}$ for $m$ steps on each solution. The proposed Bayesian Optimization resampling (BO) and Successive Halving (SH) are then used to explore important solution space based on current information and cut out unnecessary solution paths.

Our contributions are summarized below:

1. We conduct thorough experiments to show that current decision-based attacks often converge to a local optimum, thus further improvements are required.
2. Based on the idea of Bayesian optimization and successive halving, we design a meta algorithm to boost the performance of current decision-based attack algorithms and encourage them to find a much smaller adversarial perturbation efficiently.
3. Comprehensive experiments demonstrate that BOSH-attack can consistently boost existing decision-based attacks to find better examples with much smaller perturbation. In addition to the standard neural network models, we also test our algorithms on attacking discrete GBDT models and detector-based defense models. Moreover, our algorithm can reduce the computation cost by 10x compared to the naive approach.

## 2 BACKGROUND AND RELATED WORK

Given a classification model $F : \mathbb{R}^d \to \{1, \ldots, C\}$ and an example $x_0$, adversarial attacks aim to find the adversarial example that is closest to $x_0$. For example, an untargeted attack aims to find the minimum perturbation to change the predicted class, which corresponds to the following optimization problem:

$$\min_{\delta} \|\delta\| \quad \text{s.t.} \quad F(x_0 + \delta) \neq F(x_0). \tag{1}$$

Exactly minimizing (1) is usually intractable; therefore, we can only expect to get a feasible solution of (1) while hoping $\|\delta\|$ to be as small as possible.

**White-box Attack.** For neural networks, we can replace the constraint in (1) by a loss function defined on the logit layer output, leading to a continuous optimization problem which can be solved by gradient-based optimizers. This approach has been used in popular methods such as FGSM (Goodfellow et al., 2014), C&W attack (Carlini & Wagner, 2017b) and PGD attack (Madry et al., 2017). All these white-box attacks developed for neural networks assume the existence of the gradient. However, for models with discrete components such as GBDT, the objective cannot be easily defined and gradient-based white-box attacks are not applicable. There are few white-box attacks developed for specific discrete models, such as Mixed Integer Linear Programming (MILP) approach for attacking tree ensemble (Kantchelian et al., 2016). However, those algorithms are time consuming and require significant efforts for developing each model.

**Soft-label Black-box Attack** Black box setting considers the cases when an attacker has no direct access to the model's parameter and architecture, and the only valid operation is to query input examples and get the corresponding model output. In the **soft-label** black box setting, it is assumed that the model outputs the probability of each label for an input query. Chen et al. (2017b) showed that the attack can still be formulated as an optimization problem where the objective function value can be computed while the gradient is unavailable. Based on this, various zeroth order optimization algorithms have been proposed, including NES-attack (Ilyas et al., 2018a), EAD attack (Chen et al., 2018), bandit-attack (Ilyas et al., 2018b), Autozoom (Tu et al., 2019), Genetic algorithm (Alzantot et al., 2018).

**Hard-label Black-box attack (Decision-based attack)** In this paper, we focus on the hard-label black box attack (also known as decision-based attack). In contrast to the soft-label setting, the

attacker can only query the model and get the **top-1 predicted label** without any probability information. To minimize the perturbation in the decision-based setting, Brendel et al. (2017) first proposed a Boundary Attack based on random walk on the decision surface. Later on, Cheng et al. (2018) showed that hard-label attack can be reformulated as another continuous optimization problem, and zeroth order optimization algorithms such as NES can be used to solve this problem. Cheng et al. (2019) further reduces the queries by only calculating the sign of ZOO updating function, and Chen et al. (2019b) proposed another algorithm to improve over boundary attack. Such methods can converge quickly but suffer from local optimum problems, and thus require more careful and thorough search in the solution space.

**Probability based black-box optimization**  There are two commonly used methods to solve a black-box or non-differentiable optimization problem: gradient-based and probabilistic-based algorithms. Gradient-based methods are based on iterative local updates until convergence, while probabilistic-based algorithms such as Bayesian optimization (BO) (Pelikan et al., 1999; Snoek et al., 2012) approximate the objective function by a probabilistic model. Generally speaking, gradient-based methods are commonly used in black-box attack because it converges fast. However, these methods often stuck in some local optimal directions, especially when the searching space is high dimensional and non-convex. Probabilistic-based algorithms are frequently used in low-dimensional problems such as hyperparameter tuning and can have a better chance to find more global optimal values (Snoek et al., 2012; Bergstra et al., 2011). However, the computation cost grows exponentially while the dimension increases and quickly become unacceptable. Therefore, they cannot be directly applied to generate adversarial examples. In this paper, we attempt to combine Bayesian Optimization and iterative local updates to improve the solution quality of current attack algorithms while being able to scale to high dimensional problems.

**Combinatorial heuristic and genetic algorithms**  There exist various heuristic algorithms commonly applied to combinatorial optimization problems. In these algorithms, they try to leverage the effects between greedy and random. Commonly, they will search for different directions and drop the bad ones, and then put more attention on the relatively good candidates. Greedy randomized adaptive search (Feo & Resende, 1995) finds good solutions in an iterative way. It first generates a set of solutions and use a greedy function to rank these solutions. Later, good candidates are placed in a restricted candidate list, and randomly chosen when forming the solution. Tabu search (Glover & Laguna, 1999) selects a candidate and checks its immediate neighbors, trying to find an improved solution. In order to avoid stucking in local optimal areas, it maintains a list called tabu list to store past good solutions (often local optimal solutions). In further searches, it will prevent from looking for this areas. Other approaches like Genetic Algorithms (GA) and Simulated Annealing (SA) also try to adopt random in the searching process. In this paper, we simply use Successive Halving (SH) (Jamieson & Talwalkar, 2016) to remove unimportant candidate configurations iteratively. The details can be found in Section 3.1.

## 3 THE PROPOSED ALGORITHM

**Observation: Decision based attacks are easily stuck in local optimum.**  Most existing adversarial attacks adopt **iterative local updates** to find adversarial examples – starting from an initial point, they iteratively update the solution until convergence. For example, in white-box attacks such as C&W and PGD methods, they aim at optimizing a non-convex loss function by iterative gradient updates. In Figure 1(a), we plot the 2-dimensional projection of the decision surface of a neural network. We can observe that there are two local minimums[1]. Results show that there are two local minimums and the attack algorithm converges to one of them according the the initialization region. Similarly, in decision-based attacks, existing methods start from some point on the decision surface and then iteratively update the point locally on the surface either by gradient update (Cheng et al., 2018; Chen et al., 2019b) or random walk (Brendel et al., 2017). In Figure 1(b) we plot the decision surface of a GBDT. We observe a similar issue that there are many local minima in the GBDT decision boundary.

---

[1]Here local minimum indicates a point on the decision boundary that has shortest distance to the original example, compared to other nearby points on the decision boundary. Those local minimums are the points where a decision-based attack can converge to.

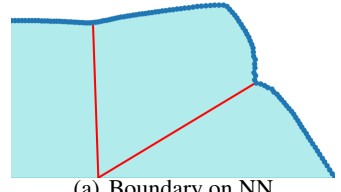

(a) Boundary on NN.

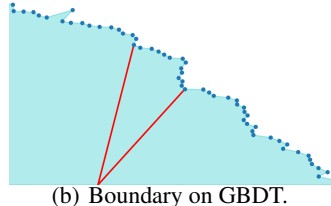

(b) Boundary on GBDT.

Figure 1: Decision boundary (NN and GBDT models on MNIST dataset) projected on a two-dimensional hyperplane. To choose which 2D hyperplane to project to, we run a decision-based attack from two random initialization points, and use their converged perturbation directions as the vector to form the hyperplane. We then query the decision boundary on this hyperplane to plot these figures.

We further quantify how serious the problem is. On an MNIST network, Figure 2(a) shows the distribution of converged adversarial perturbations of C&W attack (white-box attack) and Sign-OPT attack (decision-based attack) under approximately 400 random initial points. We observe that the converged solutions of C&W attack are quite concentrated between $[1.41, 1.47]$. However, when considering decision-based attack such as Sign-OPT, the converged solutions are widely spread from 1.36 to 1.55. In general, our experiments suggested that **decision based attacks are much more sensitive to initialization**. This is because they only update solutions on the decision boundary while C&W and PGD attack can update solution inside/outside the boundary.

Furthermore, such phenomenon is obvious when the victim model is GBDT. For example, in Figure 2(b) we can see the converged solution spread from 0.5 to 1.5 when applying Sign-OPT attack. Therefore, the solution of any single run of Sign-OPT on GBDT cannot really reflect the minimum adversarial perturbation of the given model, and thus it is crucial to design an algorithm that converges to a better solution. Since the phenomenon is more severe for decision based attacks, we will mainly focus on improving the quality of decision based attacks in this paper, while in general our method can also be used to boost the performance of white-box attacks marginally, as illustrated in Appendix A.

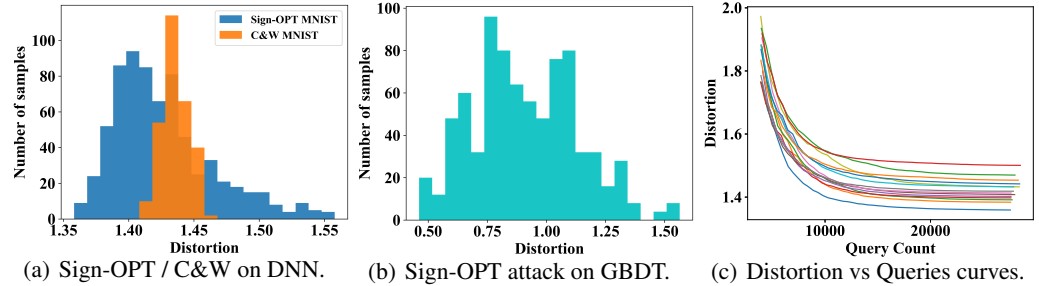

(a) Sign-OPT / C&W on DNN.   (b) Sign-OPT attack on GBDT.   (c) Distortion vs Queries curves.

Figure 2: Distribution of converged adversarial perturbation norm on an MNIST image. The figures show that the final $L_2$ distortion can be very different because of various starting directions. Figure 2(a) and 2(b) show the histogram for final perturbations, and Figure 2(c) shows the converging curve of Sign-OPT attack on a neural network model on MNIST dataset.

## 3.1 A GENERAL MECHANISM FOR IMPROVED DECISION BASED ATTACK

Given a local update based attack $\mathcal{A}$, our goal is to find a solution with improved quality. To this end, we propose a meta algorithm to address this issue by integrating probability-based (Bayesian) black-box optimization with iterative local updates. As shown in Algorithm 1, our algorithm maintains a candidate pool $\mathbb{P}_a$ that stores all the active configurations, where each configuration $\boldsymbol{u} \in \mathbb{P}_a$ is an intermediate iterate of algorithm $\mathcal{A}$. Also, we assume that there is an attack objective $\mathcal{C}$ such that $\mathcal{C}(\boldsymbol{u})$ measures the quality of the solution. For decision based attacks, the goal is to find the optimal direction to minimize the distance to the boundary along that direction (Cheng et al., 2018).

Therefore, $\boldsymbol{u}$ is the direction of adversarial perturbation and

$$\mathcal{C}(\boldsymbol{u}) = \min_{\lambda > 0} \lambda \quad \text{s.t.} \quad f\left(x_0 + \lambda \frac{\boldsymbol{u}}{\|\boldsymbol{u}\|}\right) \neq y_0,$$

where $y_0$ is the correct label. This can be computed by a fine-grained plus binary search procedure (see Cheng et al. (2018)), and in fact, in most of the algorithms $\mathcal{C}(\boldsymbol{u})$ is directly maintained during the optimization procedure (Brendel et al., 2017; Cheng et al., 2019).[2] At each iteration, we run $m$ iterations of $\mathcal{A}$ on each active configuration $\boldsymbol{u} \in \mathbb{P}_a$ to get the improved configurations. Then we conduct the following two operations to reduce the candidate pool size and to resample new configurations to explore important subspace based on Bayesian optimization. We discuss each step in details as below.

**Successive Halving (SH) to cut unimportant candidate configuration.** After updating each candidate by $m$ iterations, we compute the objective function value of each candidate and discard the worst half of them. Iteratively reducing the candidate set into half accelerates the algorithm, while still maintaining an accurate solution pool. This idea has been used in hyperparameter search (Jamieson & Talwalkar, 2016) but has not been used in adversarial attack.

**Bayesian Optimization (BO) for Guided Resampling.** To introduce variance in the intermediate steps and explore other important region, we propose a guided resampling strategy to refine the candidate pool. The general idea is to resample from the solution space in the middle step based on the knowledge acquired before and focus on promising subareas. Specifically, we use a Bayesian optimization method called Tree Parzen Estimator (TPE) (Bergstra et al., 2011) to resample new configurations.

In order to do resampling, we maintain another pool $\mathbb{P}_s$ that stores all the previous iterations performed including the cutted ones, since all the information will be useful for resampling. As shown in Algorithm 2, we first divide the observed data in $\mathbb{P}_s$ into worse and better parts based on the associated objective function value. We then train two separate Kernel Density Estimators (KDE) denoted as $l(\cdot)$ and $g(\cdot)$ on these two subsets.

$$\begin{cases} l(\boldsymbol{u}) = p(\mathcal{C}(\boldsymbol{u}) \leq \alpha | \boldsymbol{u}, \mathbb{P}_s), \\ g(\boldsymbol{u}) = p(\mathcal{C}(\boldsymbol{u}) > \alpha | \boldsymbol{u}, \mathbb{P}_s). \end{cases} \tag{2}$$

The parameter $\alpha$ is set to 20%, which ensures the better part $l(\boldsymbol{u})$ has 20% of configurations in $\mathbb{P}_s$ and the worse part $g(\boldsymbol{u})$ has the remaining 80%, relatively. Later, we sample new data with the minimum value of $l(\cdot)/g(\cdot)$, which can be proved to have maximum relative improvement in Equation 4 (see more information in Appendix B). Since we can not directly find such points, we sample for a few times (the number is set to 100 during the experiment) from $l(\cdot)$ and keep the one with the minimal $l(\cdot)/g(\cdot)$.

---

[2]When combining our method with white-box attacks, $\boldsymbol{u}$ will be a $d$-dimensional vector in the input space, and $\mathcal{C}(\boldsymbol{u})$ will be the objective defined in C&W or PGD attack.

---

**Algorithm 1** The proposed BOSH attack framework.

---

**Input:** Model $f$, original example $x_0$, attack objective $\mathcal{C}$, gradient-based attack algorithm $\mathcal{A}$, cutting interval $M$, cutting rate $s$, cutting interval increase rate $m$.

1: Randomly sample $k$ initial configurations to form $\mathbb{P}_a$ (Gaussian or uniform random).
2: $\mathbb{P}_s \leftarrow \mathbb{P}_a$.
3: **for** $t = 1, 2, \ldots$ **do**
4:     **for** each $\boldsymbol{u}_t \in \mathbb{P}_a$ **do**         // perform attack on all configurations
5:         **for** $j = 1, \ldots, M$ **do**         // conduct $M$ iterations before cutting
6:             $\boldsymbol{u}'_t \leftarrow \mathcal{A}(\boldsymbol{u}_t)$
7:             $\mathbb{P}_s \leftarrow \mathbb{P}_s \cup \{(\boldsymbol{u}'_t, \mathcal{C}(\boldsymbol{u}'_t))\}$.     // Record all interval steps
8:         Update $\boldsymbol{u}_t$ in $\mathbb{P}_a$ with $\boldsymbol{u}'_t$.     // Update the configuration in $\mathbb{P}_a$
9:     Delete the worst $s\%$ of configurations from $\mathbb{P}_a$
10:     $\mathbb{P}_a \leftarrow \mathbb{P}_a \cup$ TPE-resampling($\mathbb{P}_s, |\mathbb{P}_a| * s\%$)
11:     $M \leftarrow M \cdot (1 + m)\%$. // Increase the searching interval

---

---

**Algorithm 2** Tree Parzen Estimator resampling.

---

**Input:** Observed datas $\mathbb{P}_s$, resample times $T$;
 1: Initialize $\mathbb{P}_l$ as an empty list;
 2: Divide $\boldsymbol{u} \in \mathbb{P}_s$ into two subset $\mathbb{L}$ (better) and $\mathbb{H}$ (worse) based on objective function $\mathcal{C}$;
 3: Build two separate KDEs on $\mathbb{L}$ and $\mathbb{H}$ denoted as $l(\cdot)$ and $h(\cdot)$ respectively;
 4: Use Grid Search to find the best KDE bandwidth $b_l$, $b_h$ for $l(\cdot)$ and $h(\cdot)$;
 5: **for** each $t \in [0, T]$ **do**
 6:     initialization: $k = 0$, min_score $= \infty$;
 7:     **while** $k <$ max sample times **do**
 8:         Sample $\boldsymbol{u}_{tk}$ from $l(\cdot)$;
 9:         **if** min_score $> g(\boldsymbol{u}_{tk})/l(\boldsymbol{u}_{tk})$ **then**;
10:             $\boldsymbol{u}_t \leftarrow \boldsymbol{u}_{tk}$;
11:             min_score $= g(\boldsymbol{u}_{tk})/l(\boldsymbol{u}_{tk})$;
12:         $k \leftarrow k + 1$
13:     $\mathbb{P}_l \leftarrow \mathbb{P}_l \cup \{(\boldsymbol{u}_t, \mathcal{C}(\boldsymbol{u}_t))\}$;
14: **return** $\mathbb{P}_l$;

---

The reason we use TPE for resampling is that the computational cost grows linearly with the number of data points in $\mathbb{P}_s$. In comparison, traditional Bayesian optimization method like Gaussian Process (GP) will require cubic-time to generate new points. Therefore, TPE is more suitable for high dimensional problems.

In the experiments we find that the final best configuration mostly comes from resampling, instead of the set of starting configurations. This proves the effectiveness of resampling during search, the quantitative results will be shown in Section 4.2.

## 4 EXPERIMENTS

We conduct experiments on various models and datasets to verify the efficiency and effectiveness of the proposed approach. We try to enhance the performance of decision-based attack on image classification tasks like MNIST, CIFAR-10 and ImageNet, and also conduct experiments on tree model like GBDT and detection model like LID. Furthermore, we demonstrate that our meta-algorithm is also able to improve existing white-box attacks.

### 4.1 DECISION-BASED ATTACK ON NEURAL NETWORKS

We conduct experiments on three standard datasets: MNIST (LeCun et al., 1998), CIFAR-10 (Krizhevsky et al., 2010) and ImageNet-1000 (Deng et al., 2009). The neural network model architecture is the same with the ones reported in Cheng et al. (2018): for both MNIST and CIFAR we use the network with four convolution layers, two max-pooling layers and two fully-connected layers, which achieve 99.5% accuracy on MNIST and 82.5% accuracy on CIFAR-10 as reported in (Carlini & Wagner, 2017b; Cheng et al., 2018). For ImageNet, we use the pretrained Resnet-50 (He et al., 2016) network provided by torchvision (Marcel & Rodriguez, 2010), which achieves a Top-1 accuracy of 76.15%. We randomly select 100 examples from test sets for evaluation. The parameters of the proposed algorithms can be found in Table 6 in Appendix D.

**Improved solution quality of existing methods**    We compare the solution quality of the proposed algorithm with three existing decision-based attack methods: Boundary attack (Brendel et al., 2017), OPT-attack (Cheng et al., 2018) and Sign-OPT attack (Cheng et al., 2019) on MNIST, CIFAR-10 and ImageNet data sets. For our algorithm, we use Sign-OPT attack as the base algorithm and set $k = 30$ for the initial candidate pool. The average $L_2$ perturbation of our method and baselines are presented in Table 8. Note that all the decision based attacks maintain intermediate iterates on the decision boundary, so they always output a successful attack. The main comparison is the average $L_2$ perturbation to alter the predictions. We also follow Cheng et al. (2018) to report Attack Success Rate (ASR) by calculating ratio of adversarial examples with perturbation $< \epsilon$ ($\epsilon$ is chosen based on

different tasks). The results show that can help decision-based attacks achieve lower $L_2$ perturbation and higher attack success rate. The detailed analysis is shown in the next section.

The proposed algorithm can also be used to boost the performance of other decision based attacks. Table 7 in the Appendix demonstrates that the proposed algorithm consistently improves the $L_2$ perturbation and success rate of Boundary attack and OPT-attack.

Table 1: Results of hard-label black-box attack on MNIST, CIFAR-10 and ImageNet-1000. We compare the performance of several attack algorithms under untargeted setting.

|  | **MNIST** | | **CIFAR-10** | | **ImageNet-1000** | |
| --- | --- | --- | --- | --- | --- | --- |
|  | Avg $L_2$ | ASR ($\epsilon < 1.0$) | Avg $L_2$ | ASR ($\epsilon < 0.13$) | Avg. $L_2$ | ASR ($\epsilon < 1.4$) |
| C&W (White-box) | 0.96 | 60% | 0.12 | 61% | 1.53 | 49% |
| Boundary attack | 1.13 | 41% | 0.15 | 48% | 2.02 | 19% |
| OPT-based attack | 1.09 | 46% | 0.14 | 50% | 1.67 | 38% |
| Sign-OPT attack | 1.07 | 49% | 0.14 | 51% | 1.43 | 59% |
| **BOSH Sign-OPT attack** | **0.91** | **67%** | **0.10** | **65%** | **1.18** | **81%** |

## 4.2 ANALYSIS

We then conduct a study to test each component of our algorithm and compare with the baselines. The experiment is done on MNIST data using Sign-OPT attack as the base attack method. The results are summarized in Table 2.

Table 2: Comparions for the effectiveness of successive halving and TPE resampling. The Relative Gain is based on Multi-Directional attack and Queries means total queries of all directions.

|  | Starting Directions | Avg $L_2$ | Ratio | ASR ($\epsilon < 1.0$) | Queries | Ratio |
| --- | --- | --- | --- | --- | --- | --- |
| Multi-initial Sign-OPT | 1 | 1.07 | 0% | 49% | 25,456 | 1x |
|  | 30 | 0.98 | 8.4% | 57% | 771,283 | 30x |
|  | 50 | 0.94 | 12.1% | 63% | 1,268,392 | 50x |
|  | 100 | 0.91 | 15.0% | 65% | 2,567,382 | 100x |
| Successive Halving Sign-OPT | 30 | 0.99 | 7.5% | 55% | 161,183 | 6.3x |
| **BOSH Sign-OPT** | **30** | **0.91** | **15.0%** | **67%** | **252,014** | **9.9x** |
|  | **50** | **0.88** | **17.8%** | **71%** | **409,841** | **16.1x** |
|  | **100** | **0.87** | **18.7%** | **71%** | **829,865** | **32.6x** |

**Comparison with naive mulitple initialization approach.** A naive way to improve the solution quality of existing attack is to run the attack on multiple random initialization points. This strategy has been used in white-box attack[3] and is also applicable to the decision based attacks. We compare Sign-OPT with 30, 50, 100 initial points and the proposed BOSH boosted Sign-OPT approach in Table 2. The results demonstrate that successive halving requires much less queries than naively running multiple initial configurations. Due to resampling, the proposed approach converges to a better solution under the same initial pool size. For example, to achieve average $0.91$ $L_2$ perturbation, BOSH boosted Sign-OPT requires 10 times less queries than multi-initial Sign-OPT.

**Size of the initial pool.** The size of initial pool (denoted by $k$ in our algorithm) is an important parameter. Table 2 shows that increasing $k$ only has a marginal effect after $k \approx 30$. When introducing cutting and resampling mechanism into the Sign-OPT attack, the final best perturbation is less sensitive to the number of starting directions, which means that resampling tend to make the search less dependent on the starting directions. Detailed discussion is in Appendix C.

---

[3]See the leaderboard at `https://github.com/MadryLab/mnist_challenge`

**Effect of successive halving and TPE resampling.** We study the effect of these two components separately. As shown in Figure 3(a), the approach of successive halving keeps throwing away the worse $s$ percent of configurations until converge during a specific interval until there is only one sample left. When combining this with resampling, as in Figure 3(b), our algorithm finds directions that are better than original ones. We observed emperically that the final best direction often comes from resampling instead of the original starting directions. This observation demonstrates the importance of resampling in the intermediate steps. Furthermore, Table 2 shows that combining Sign-OPT with successive halving (second column) has worse solutions compared with BOSH Sign-OPT. This indicates that resampling is important for getting a better solution.

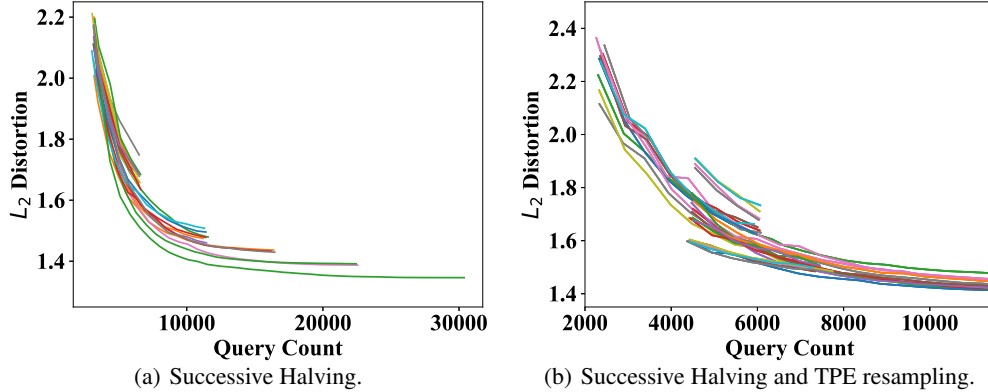

(a) Successive Halving.      (b) Successive Halving and TPE resampling.

Figure 3: Illustration of the effect of Successive Halving and TPE resampling. Note that Figure 3(b) only exhibits **part of the curve** to show the effect of TPE.

**What is the best cutting interval?** The parameter $M$ decides how many iterations are applied using base attacker before the next cutting/resampling stage. This is an important parameter to be tuned. If $M$ is too small, some solution paths will be wrongly throwing away; while if $M$ is too large, the whole procedure requires a large number of queries. In our experiment, we use a subset of images to tune this parameter and find that the images in the same dataset often share similar best cutting interval. This reduces lots of unnecessary computations. The parameters for different datasets are shown in Appendix D.

## 4.3 DECISION-BASED ATTACK ON OTHER MODELS

We conduct untargeted attack on gradient boosting decision tree (GBDT). Since Sign-OPT does not include the experiment with GBDT, we use the OPT-based attack (Cheng et al., 2018) and apply our meta algorithm on top of it. We consider two datasets, MNIST and HIGGS, and use the same models provided by (Cheng et al., 2018).[4]

We compare the average $L_2$ perturbation and the attack success rate in Table 3. The results show that the proposed method significantly boosts the performance of OPT attack. The overall improvement is more significant than attacking neural networks. This is mainly because that the decision boundary of GBDT contains more local minima than neural networks, as plotted in Figure 1.

Table 3: Comparison of results of untargeted attack on gradient boosting decision tree.

|  | HIGGS | | MNIST | |
|---|---|---|---|---|
|  | Avg $L_2$ | ASR ($\epsilon < 0.15$) | Avg $L_2$ | ASR ($\epsilon < 0.8$) |
| OPT-based attack | 0.169 | 52% | 0.952 | 49% |
| TPE-SH attack | 0.103 | 81% | 0.722 | 79% |

Table 4: Results of attack on MNIST detector models under untargeted setting.

|  | Avg $L_2$ | ASR ($\epsilon < 1.5$) |
|---|---|---|
| CW HC attack | 2.14 | 20.25% |
| Sign-OPT attack | 1.24 | 52.63% |
| BOSH attack | 1.18 | 71.42% |

---

[4] The MNIST model is downloaded from LightGBM and use the parameters in `https://github.com/Koziev/MNIST_Boosting`, which achieves 98.09% accuracy. For HIGGS, we can achieve 0.8457 accuracy relatively.

### 4.3.1 DECISION-BASED ATTACK ON DETECTION MODELS

To improve the performance of neural networks, a line of research, such as KD+BU (Feinman et al., 2017), LID (Ma et al., 2018), Mahalanobis (Lee et al., 2018) and ML-LOO (Yang et al., 2019), has been focusing on screening out adversarial examples in the test stage without touching the training of the original model. Besides comprehensive evaluation of our attack on various classification models with a variety of data sets, we carry out experimental analysis of our untargeted attack on one state-of-the-art detection model LID (Ma et al., 2018) on MNIST data set. To train a detection model on MNIST, we first train a simple classification network composed of two convolutional layers followed by a hidden dense layer with 1024 units. Then we apply C&W attack to this model to generate adversarial examples from the original test samples. Finally we train LID detectors with the original test samples and adversarial examples we have generated with the standard train/test split. The state-of-the-art detection model LID achieves 0.99 test accuracy.

C&W high confidence attack (Carlini & Wagner, 2017a) has been shown to have great performance in attacking various detection models. So we compare the average $L_2$ perturbation and attack success rate of three attacking methods C&W high confidence attack, Sign-OPT attack and BOSH Sign-OPT attack in Table 4. At each query, we define the attack to be successful if it fools both the detector model and the original model. The results show that the proposed method can significantly boost the performance of the Sign-OPT attack and it achieves much better performance than C&W high confidence attack.

## 5 CONCLUSION

In this paper, we propose a meta algorithm to boost the performance of existing decision based attacks. In particular, instead of traversing a single solution path when searching for an adversarial example, we maintain a pool of solution paths to explore important regions. We show empirically that the proposed algorithm consistently improves the solution quality of many existing decision based attacks, and can obtain adversarial examples with improved quality on not only neural networks, but also other decision based models, such as GBDT and detection-based models.

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

## A    BOOSTING WHITE-BOX ATTACK

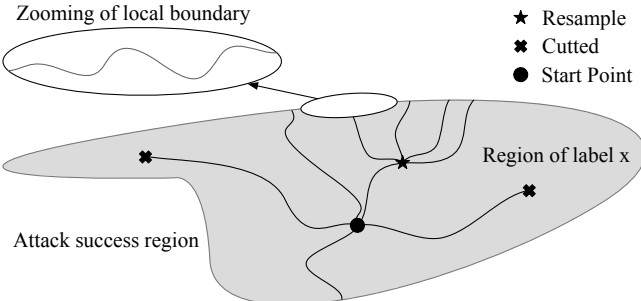

Figure 4: Illustration of a possible boundary distribution and attack steps on it. Starting from different directions, we conduct cutting and resampling during the middle steps. The directions that are not promising are cut to save computational cost, and the directions that reach lower error value will be expanded to encourage exploring. This figure also shows that the boundary can be very unsmooth and contains lots of local optimal points on the surface.

Figure 4 shows a possible boundary distribution and C&W (Carlini & Wagner, 2017b) attack performed on it. The decision boundary of a neural network can be very unsmooth and contains lots of local optimal points. Generally speaking, the cost from the original point toward the boundary(which means a successful attack) highly depends on the directions. Traditional white-box attack algorithms like FSGM (Goodfellow et al., 2014), PGD (Kurakin et al., 2016) and C&W (Carlini & Wagner, 2017b) use walk through gradient to reach the boundary, implying that gradient can guide to the optimal value, which may not be the case. We try to improve C&W attack to find a global local optimal adversarial example by encouraging it to search not just depends the gradient. Assume that the original sample is $(\boldsymbol{x}, y)$ and $L$ is the logit output of the neural network's loss function, C&W attack conducts iterative search as following:

$$
\begin{aligned}
\arg\min_{\Delta\boldsymbol{x}} \quad & \|\Delta\boldsymbol{x}\|_p + c \cdot f(\boldsymbol{x} + \Delta\boldsymbol{x}) \quad \text{s.t. } \boldsymbol{x} + \Delta\boldsymbol{x} \in [0,1]^n \\
\text{where} \quad & f(x) = \max(0, \max_{i \neq y}(L(\boldsymbol{x})_i) - L(\boldsymbol{x})_y)
\end{aligned}
\tag{3}
$$

Instead of starting from a single direction which is calculated by gradient on the original point, we randomly sample $k$ points within an $\epsilon$ ball (which means the the $L_\infty$ distance between the generated point and the original one is less than $\epsilon$, this value may be slightly different depends on datasets) as a set of possible configurations. We then run the BOSH algorithm with successive halving and resampling to iteratively refine these configurations and run $m$ steps of C&W attacks. We use the loss function in the above objective for TPE resampling.

### A.1    RESULTS

Although we focus on decision based attack throughout the paper, the proposed method can actually also boost the performance of white-box attacks. We use C&W attack (Carlini & Wagner, 2017b) as the base method and perform experiments on MNIST and CIFAR-10 datasets. In order to introduce variance and encourage the model to find better and global optimal values, we randomly sample 50 points inside the $L_2$ $\epsilon$ ball (we use $\epsilon = 0.3$ during the experiment) as the starting points, and apply C&W attack on each of them. For simplicity, we fix $c$ described in (3) to be 0.2. The results before and after applying our algorithm are shown in Table 5. We can observe that although our algorithm can also improve C&W attack, the improvements are not as significant as that in decision based attacks. This is probably because C&W attack is less sensitive to the initial point, as demonstrated in Figure 2(a).

Table 5: Comparison of results of untargeted attack on white-box attack.

|  | MNIST | | CIFAR-10 | |
|---|---|---|---|---|
|  | Avg $L_2$ | ASR ($\epsilon < 1.0$) | Avg $L_2$ | ASR ($\epsilon < 0.13$) |
| C&W attack | 0.96 | 60% | 0.12 | 61% |
| BOSH attack | 0.91 | 67% | 0.10 | 66% |

## B  BAYESIAN OPTIMIZATION AND SUCCESSIVE HALVING

### B.1  BAYESIAN OPTIMIZATION (BO)

**Bayesian Optimization (BO)** has been successfully applied to optimize a function which is non-differentiable or black-box like finding the hyper-parameters of neural networks in AutoML area. It mainly adopts the idea to sample new points based on the past knowledge. Basically, Bayesian optimization finds the optimal value of a given function $f : \mathcal{X} \to \mathbb{R}$ in a iterative manner: at each iteration i, BO uses a probabilistic model $p(f|\mathcal{D})$ to estimate and approach the unknown function $f$ based on the data points that are already observed by the last iterations. Specifically, it samples new data points $x_t = \text{argmax}_{\boldsymbol{x}} u(\boldsymbol{x}|\mathcal{D}_{1:t-1})$ where $u$ is the acquisition function and $\mathcal{D}_{1:t-1} = \{(\mathbf{x}_1, y_1), \ldots, (\mathbf{x}_{t-1}, y_{t-1})\}$ are the $t-1$ samples queried from $f$ so far. The most widely used acquisition functions is the expected improvement (EI):

$$\text{EI}(\boldsymbol{x}) = \mathbb{E}_{\text{x} \sim p} \max(f(x) - f(x^+), 0) \tag{4}$$

Where $f(\boldsymbol{x}^+)$ is the value of the best sample generated so far and $\boldsymbol{x}^+$ is the location of that sample, i.e. $\boldsymbol{x}^+ = \arg\max_{\boldsymbol{x}_i \in \mathcal{D}} f(\boldsymbol{x}_i)$.

### B.2  THE TREE PARZEN ESTIMATOR (TPE).

**The Tree Parzen Estimator (TPE).** TPE (Bergstra et al., 2011) is a Bayesian optimization method proposed to solve the hyper-parameter tuning problems that uses a kernel density estimator (KDE) to approximate the distribution of $\mathcal{D}$ instead of trying to model the objective function $f$ directly. Specifically, it models the $p(x|y)$ and $p(y)$ instead of $p(y|x)$, and define $p(x|y)$ using two separate KDE $l(x)$ and $g(x)$:

$$p(x|y) = \begin{cases} l(x) & y \leq \alpha \\ g(x) & y > \alpha \end{cases} \tag{5}$$

where $\alpha$ is a constant between the lowest and largest value of $y$ in $\mathcal{D}$. Bergstra et al. (2011) shows that maximizing the radio $l(\boldsymbol{x})/g(\boldsymbol{x})$ is equivalent to optimizing the EI function described in Equation 4 (see Theorem 1 for more detail). In such setting, the computational cost of generating a new data point by KDE grows linearly with the number of data points already generated, while traditional Gaussian Process (GP) will require cubic-time.

**Theorem 1** *In Equation 5, maximizing the radio $l(x)/g(x)$ is equal to optimizing the Expected Improvement (EI) in Equation 4*

**Proof:**

The Expected Improvement can also be written as:

$$\text{EI}(\boldsymbol{x}) = \mathbb{E}_{\text{x} \sim p} \max(f(x) - f(x^+), 0) = \int_{-\infty}^{\alpha} (y^* - y) p(y|x) dy = \int_{-\infty}^{\alpha} (\alpha - y) \frac{p(x|y)p(y)}{p(x)} dy \tag{6}$$

Assume that $\gamma = p(y < \alpha)$, then:

$$p(x) = \int_{\mathbb{R}} p(x|y)p(y) dy = \gamma l(x) + (1 - \gamma) g(x) \tag{7}$$

Therefore,

$$\int_{-\infty}^{\alpha} (\alpha - y) p(x|y) p(y) dy = l(x) \int_{-\infty}^{\alpha} (\alpha - y) p(y) dy = \gamma \alpha l(x) - l(x) \int_{-\infty}^{\alpha} p(y) dy \tag{8}$$

So finally,

$$EI_\alpha(x) = \frac{\gamma\alpha l(x) - l(x)\int_{-\infty}^{\alpha} p(y)dy}{\gamma l(x) + (1-\gamma)g(x)} \propto (\gamma + \frac{g(x)}{l(x)}(1-\gamma))^{-1} \tag{9}$$

Which means maximizing $l(x)/g(x)$ is equivalent to maximize the EI function.

### B.3 SUCCESSIVE HALVING.

**Successive Halving.** The idea behind Successive Halving (Jamieson & Talwalkar, 2016) can be easily illustrated by it's name: first initialize a set of configurations and perform some calculations on them, then evaluate the performance of all configurations and discard the worst half od these configurations, this process continues until there is only one configuration left. BOHB (Falkner et al., 2018) combines HyperBand (derived from Successive Halving) (Li et al., 2016) and TPE to solve the AutoML problem and achieve greate success.

However, these methods are originally applied to the hyper-parameter tuning problem where the parameters need to be searched are not too much(approximately 10-20), it will suffer from "dimensional curse" when the number of parameters grows larger and the computation cost needed will be unacceptable. There are already some work (Moriconi et al., 2019; Wang et al., 2013) try to use BO in high dimension, while we still found in experiment that simply use BO can not converge as good as gradient-based methods.

## C DISCUSSION ABOUT THE NUMBER OF STARTING DIRECTIONS

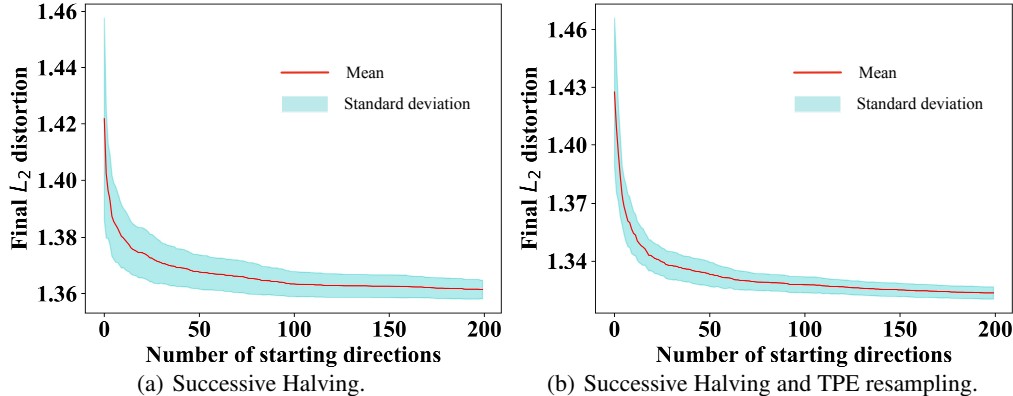

(a) Successive Halving.  (b) Successive Halving and TPE resampling.

Figure 5: Number of starting directions vs Final $L_2$ distortion. Since we will run multiple times on each setup to reduce variance, the red line shows the average distortion and the blue area shows the standard deviation.

We discuss the problem that how many starting points are enough for a successful attack. In order to find the best number of starting points, we conduct attack on an image with different number of starting directions, for a specific number of starting directions, we also run several times and average the result to reduce variance. Figure 5 shows the attack on MNIST image using Sign-OPT method, we can see the effect that the number of starting direction have on the final converging perturbation. Also, We can find that the standard deviation is smaller and the final perturbation is lower when resampling by TPE is introduced. This is probably because TPE resampling also introduce variance in the middle step and making the algorithm not completely depends on the starting directions, this will also helps increase the probability to find a better optimal value .

## D  PARAMETERS OR DIFFERENT DATASETS

Table 6: Parameters for Sign-OPT attack on different image classification datasets.

| Dataset | Maximum Queries Per Direction | Cutting and Resampling Interval | Interval Increase Ratio |
|---|---|---|---|
| MNIST | 40000 | 3500 | 1.4 |
| CIFAR-10 | 20000 | 2000 | 1.3 |
| ImageNet | 200000 | 6000 | 1.6 |

## E  BOOSTING DECISION-BASED ATTACK ALGORITHMS

To demonstrate that our algorithm can consistently boost existing hard-label attack algorithms, we try to enhance the performance of three decision-based algorithms described in Section 4.1. All the parameters are equal to Section 4.1 and we use 30 starting directions for our boosting algorithm. The results are shown in Table 7.

Table 7: Results of hard-label black-box attack on MNIST, CIFAR-10 and ImageNet-1000. We compare the performance several attack algorithms under untargeted setting.

| | MNIST | | | CIFAR-10 | | |
|---|---|---|---|---|---|---|
| | Avg $L_2$ | ASR ($\epsilon < 1.0$) | Queries | Avg $L_2$ | ASR ($\epsilon < 0.13$) | Queries |
| Boundary attack | 1.13 | 41% | 157,323 | 0.15 | 48% | 212,093 |
| BOSH Boundary attack | 0.99 | 53% | 1,673,837 | 0.12 | 55% | 2,239,488 |
| OPT-based attack | 1.09 | 46% | 91,834 | 0.14 | 50% | 142,498 |
| BOSH OPT-based attack | 0.95 | 62% | 983,283 | 0.11 | 62% | 1,527,384 |
| Sign-OPT attack | 1.05 | 51% | 25,456 | 0.14 | 51% | 15,285 |
| BOSH Sign-OPT attack | 0.91 | 67% | 252,014 | 0.10 | 65% | 142,738 |

## F  RUN-TIME COMPARISON

We evaluate the efficiency of various algorithms based on the number of queries, it is commonly used in the papers of this area. In this section we also include the run-time performance comparison of these methods. We use one Nvidia GTX 1080 Ti to conduct the experiments, but the run-time will reduce if we use multiple GPUs since our boosting algorithm can be easily parallelize (searches with different directions do not depend on each other).

Table 8: Results of hard-label black-box attack on MNIST and CIFAR-10. We compare the performance based on run-time.

| | MNIST | | CIFAR-10 | |
|---|---|---|---|---|
| Avg. $L_2$ | Avg $L_2$ Run-time | Run-time | Avg $L_2$ | Run-time |
| Boundary attack | 1.13 | 346.12$s$ | 0.15 | 524.92$s$ |
| OPT-based attack | 1.09 | 205.27$s$ | 0.14 | 351.59$s$ |
| Sign-OPT attack | 1.05 | 56.24$s$ | 0.14 | 37.13$s$ |
| **BOSH Sign-OPT attack** | **0.91** | **671.32$s$** | **0.10** | **462.19$s$** |

## G  ATTACK SUCCESS RATE UNDER DIFFERENT PERTURBATION $\epsilon$

In this section, we show the results of Sign-OPT and Boosted Sign-OPT attack on MNIST and CIFAR-10 dataset. We mainly show how Attack Success Rate (ASR) changes based on different perturbations.

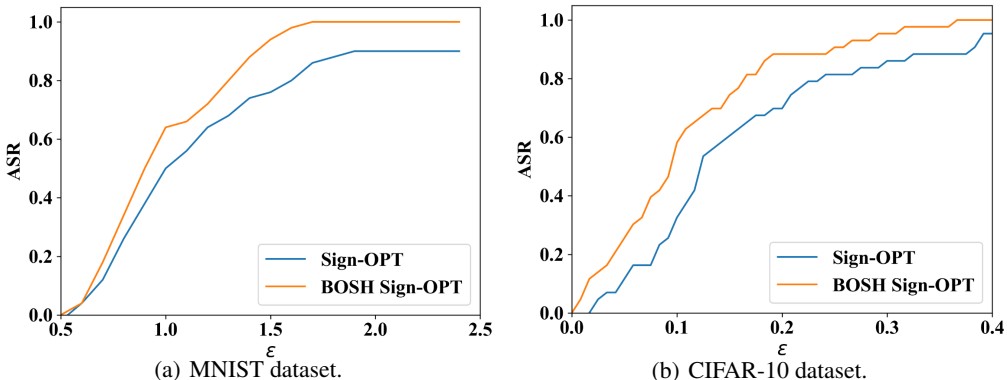

(a) MNIST dataset.              (b) CIFAR-10 dataset.

Figure 6: Perturbation $\epsilon$ vs Attack Success Rate (ASR).

## H  TIME COMPLEXITY ANALYSIS

We briefly analyse the number of queries our algorithm need regarding the parameters we mentioned in Algorithm 1. Generally speaking, in first several cutting intervals (we consistently set to 3 in the experiments), our boosting algorithm requires about k times more queries than single search. This is because we resample new configurations while cutting unpromising one. After this, we only cut and do not resample, and run until there is only one configuration left and it converges.

As discussed in Algorithm 1, assume that the cutting interval is $M$, cutting rate is $s$, cutting interval increase rate is $m$ and initial number of starting configurations is $k$.

In the cutting and resampling phase, since we only resample 3 times, we need:

$$k * (M + M(1 + m) + M(1 + m)^2) \tag{10}$$

queries. After that, we only cut the unpromising configurations, so we need:

$$k * M(1 + m)^2 * (1 + s\% + s^2\% + ...) \tag{11}$$

queries.

We can see that the main differences between the original and our algorithm is the initial number of staring configurations $k$, and we also discuss how $k$ influences the results in Section 4.2 and Appendix B.

