# OpenReview forum: "BOSH: An Efficient Meta Algorithm for Decision-based Attacks"
_ICLR.cc/2020/Conference — Reject_

### Official Review · AnonReviewer2 · 2019-10-15
**Official Blind Review #1547**

**Rating:** 3

**Review:**

In this paper, the authors study the adversarial example generation problem, in the difficult case where the attacked model is a black box. Since the model is unknown, the approaches based on the minimization of a loss function with a gradient based optimizer do not apply. The current alternatives, known as decision-based attack, use iterative local updates from a starting point to a local minimum, where the class of the adversarial example is different from the initial example while its distance stays close to the initial one.

For handling the sensibility to starting points, the authors propose a meta-algorithm, which uses any iterative local update based attacks, and which maintains a set of solutions corresponding to different starting points. The proposed algorithm uses successive halving for iteratively maintaining empirical good solutions by discarding the worst half of solutions at each step, and uses Tree Parzen Estimator to explore by resampling promising area.

In the experiments, the meta-algorithm uses SignOPT attack. It is compared with three decision based attacks, including SignOPT. Three image datasets are used. The attacked models are neural networks and gradient boosting tree in the last experiment.

Pros:
- The paper is well-written and easy to follow.
- Generic algorithm.


Cons:
- No analysis is provided.
- The proposed algorithm has a lot of parameters: $k, M, s, m, T, inf, \alpha$.
- The $\epsilon$ are not the same for each attacked model (table 1 and 3). Is it the result of a post-optimization? Could you plot the curves ASR versus $\epsilon$ ?
- In algorithm 2, $min_score=inf$,  whatever $t$, so why using two variables ?

___________________________________________________________________________________________________________________________________
I read the rebuttal.
Thanks you for answering my concerns.

I think that it is possible to provide some theoretical guanrantees. For instance, may be one could show that the quality of the attacks is increasing when Algorithm 1 is run. Finding the highest increasing rate could be useful for tuning the parameters of the algorithm.
However, I understand that this could be tricky.

I took a look to Figure 6. Good point: BOSH Sign-OPT attack outperforms Sign-OPT attack whatever $\epsilon$.


**Experience Assessment:**

I have read many papers in this area.

**Review Assessment: Checking Correctness Of Derivations And Theory:**

N/A

**Review Assessment: Checking Correctness Of Experiments:**

I assessed the sensibility of the experiments.

**Review Assessment: Thoroughness In Paper Reading:**

N/A

---

> ### Author Response · Authors · 2019-11-15
> **Response to Reviewer 2**
>
> We thank Reviewer 2 for the valuable comments. We have addressed the suggestions. Please see the responses below.
>
> [Analysis]
> Our algorithm aims to improve the solution of a base attack, so if the base attack can converge to a stationary point, our algorithm will inherit the same property. Beyond convergence to stationary points, since finding the global minimum of adversarial perturbation is NP-hard (see Katz et al., “Reluplex: An Efficient SMT Solver for Verifying Deep Neural Networks”), it’s almost impossible to guarantee the convergence to a global minimum (point on the decision boundary with minimum distance to the original example).
>
> [a lot of parameters]
> For the completeness of the algorithm, we list all the relevant hyper-parameters. However, our algorithms do not sensitive to all of them. In fact, in the experiments, we fix parameters $k$, $s$, $T$ and $\alpha$ and only tune $M$ and $m$. Also, $inf$ just means infinity and it is not a parameter. We also provide a table discussing how we choose the parameters for different datasets in Appendix D.
>
> [$\epsilon$ different for different attack model]
> Thanks for the suggestion. We have added the curves of ASR versus $\epsilon$ in Appendix G, and the results show that BOSH Sign-OPT is consistently better than Sign-OPT. Different tasks/models have different difficulties to attack, and that is why we chose different $\epsilon$ in the table. We select a relatively good $\epsilon$ value so we can compare different methods.
>
> We also want to emphasize that the main criterion for comparing decision-based attack is the average distortion. All the decision-based attacks included in our comparisons are iterating on the decision boundary or outside decision boundary. Therefore all the iterates of these algorithms are adversarial examples, and thus it’s more important to compare their distance to the original example (Avg $L_2$). ASR is counting the ratio of adversarial examples within certain $\epsilon$, which is just a way to summarize the $L_2$ distance statistics.
>
>
> [Why two variables in algorithm 2]
> $min_score = \inf$ means set min_score to an infinity value and the variable min_score is used to store the minimum score in the following while loop. In algorithm 2, we miss a line after line 9 to update $min_score$ to currently minimum value: $min_score = g(u_{tk})/l(u_{tk})$. We have corrected this in the revision. Sorry for the mistake!

---

### Official Review · AnonReviewer1 · 2019-10-25
**Official Blind Review #1**

**Rating:** 3

**Review:**

This paper proposes a meta-algorithm for the so-called "decision-based attack" problem, where a model that can be accessed only via label queries for a given input is attacked by a minimal perturbation to the input that changes the predicted label. The algorithm, BOSH, augments any iterative algorithm for this problem with a diversification strategy based on bayesian optimization and throwing away bad solutions. Empirically, it is shown that BOSH can improve the performance of recently developed algorithms for this problem, by exploring more solutions and refining them intelligently.

Overall, the decision-based attack problem is very practically relevant as it assumes minimal access to the classifier. I also really like that the authors looked into tree-based models in addition to neural networks. The algorithmic ideas that are proposed are simple and effective, as supported by the experimental results.

However, I have some serious comments about the experimental evaluation that I believe can substantially improve the quality of the paper, if addressed. Whether I raise my score or not will depend on how well the authors address these questions. I also have concerns about related work in heuristic algorithms.

Questions:
- Related work: the ideas of diversifying solution paths and throwing away bad solutions are very popular in combinatorial heuristics. You should do a thorough review of work in that area and in Genetic Algorithms (GA). You could look into the following classical/survey papers as starting points, in particular the first survey's chapter 4.

Glover, Fred, and Manuel Laguna. "Tabu search." Handbook of combinatorial optimization. Springer, Boston, MA, 1998. 2093-2229.
Feo, Thomas A., and Mauricio GC Resende. "Greedy randomized adaptive search procedures." Journal of global optimization 6.2 (1995): 109-133.

- Other Black-Box algorithms: you should compare against well-established black-box optimization algorithms such as NOMAD and RBF-OPT. Both are based on very solid mathematical foundations and have high-quality open source implementations:
NOMAD: https://www.gerad.ca/nomad/
rbf-opt: https://github.com/coin-or/rbfopt

- Runtime comparison: your analysis with respect to number of queries is very good and insightful. In addition, we should get a sense of the runtime performance. If you run each of the approaches with the same time limit, how do they fare?

- Comparing to "optimal" attacks: we need to know how well the solutions are compared to the best possible, or a close-enough approximation. You could run white-box attacks and compare the relative error to the quality of the white-box attack. Otherwise, it is hard to tell what gap remains to be closed algorithmically and it is difficult for other researchers to know whether it's worth trying to improve what you propose here in the future.

- Time complexity: please give a time complexity analysis of BOSH as a function of all its hyperparameters.

Clarity:
- Figure 1: I don't understand what this figure shows. Where are the 2 minima? Please clarify further.
- You minimize l(.) / g(.) in Algorithm 2, but maximize it in Appendix B.2. Maximizing makes more sense. Which one is it?
- Theorem 1: Is that your result or Bergstra et al.'s?

Minor comments:
- "Adversarial example generation becomes a viable method for evaluating the robustness of a machine learning model." --> "Adversarial example generation has become a viable method for evaluating the robustness of a machine learning model."
- "when searching an adversarial" --> "when searching for an adversarial"
- "Distortion" is used in the literature much less than "Perturbation"; consider switching them.
- "our mega algorithm" --> "our meta-algorithm"
- Please use consistent notation: SignOPT or Sign-OPT.
- Appendix B.1: "undifferentiable" --> "non-differentiable"
- "are the t − x1 samples" --> "are the t − 1 samples"

**Experience Assessment:**

I have read many papers in this area.

**Review Assessment: Checking Correctness Of Derivations And Theory:**

I assessed the sensibility of the derivations and theory.

**Review Assessment: Checking Correctness Of Experiments:**

I carefully checked the experiments.

**Review Assessment: Thoroughness In Paper Reading:**

I read the paper thoroughly.

---

> ### Author Response · Authors · 2019-11-15
> **Response to Reviewer 1 # Part 1**
>
> We thank Reviewer 1 for the detailed comments and critiques.
>
> [Related work]
> We thank the reviewer for the references and we have added a new paragraph in the related work section to discuss combinatorial heuristics and Genetic Algorithms.
>
> [Other black-box optimization algorithms]
> Thank you for the suggestion. Despite there are several optimization algorithms and well-established packages for optimizing black-box functions, these approaches cannot be straightforwardly applied to derive adversarial attack due to the following reasons:
>
> 1. It’s not clear whether these methods can be used to conduct decision-based attacks.
> In the decision-based attack literature, these methods have not been applied in any previous paper. Although theoretically, with the formulation of (Cheng et al., 2019), decision-based attack can be formulated as a black-box optimization problem; in practice, none of these algorithms (NOMAD/rbf-opt) has been used due to the high dimensionality of the attack objective function. For example, on ImageNet data, the input dimension is 224 * 224 *3, leading to a black-box optimization problem with 150,528‬ variables. This is beyond the scalability of classical black-box optimization algorithms, and it’s nontrivial to apply them for decision-based attack. We do agree it will be interesting to study whether those algorithms can be applied to decision-based attack, but applying those algorithms itself will be a separate paper and is out of the scope of this study. For example, there are papers applying genetic algorithms to soft-label black-box attack, and in fact developing such method is nontrivial even for soft-label black-box settings (see Moustafa et al., “Genattack: Practical Black-box Attacks with Gradient-free Optimization”). Given that hard-label black-box (decision-based) attack is much more complex than soft-label black-box attack, we believe applying NOMAD/rbf-opt to conduct decision-based attack is nontrivial.
>
> 2. In addition, we want to emphasize that our goal in this paper is not to find a good optimizer to solve the decision-based attack objective proposed by (Cheng et al., 2019). Instead, our goal is to propose a meta-algorithm such that given an existing iterative local update-based attack (denoted by A), our algorithm can boost the performance of A by a mixed strategy of Successive Halving + TPE resampling. Therefore, even if NOMAD/RBF-OPT attack exists (which is not the case as we argued in 1), they can be viewed as a base attack denoted by A, and our meta-algorithm can be used to improve their performance, not competing with them.
>
> [Runtime comparison]
> Thank you for the suggestion. We have added the runtime comparison in Appendix F. In fact, our method is a meta-algorithm on top of a base attack, so it cannot be faster than running the base attack. However, we can get much better solutions than the base attack, where the base algorithm, even if it runs for a very long time, it cannot converge to such good solution (we stop each attack when their solution converged in Table 8). Furthermore, for decision-based attacks, people mostly care about the number of queries. Imagine we are attacking a Google Cloud image recognition system, then only the number of queries is important since Google Cloud will limit the number of queries for each user, but the computation can be done off-line using multiple servers.
>
> [Comparing to "optimal" attacks]
>
> In table 1, we do include the results of C&W attack in comparison and note that it is recognized as one of the best white-box attack methods. However, white-box attacks may not always outperform the proposed decision-based attack. This is because white-box attacks are solving a non-convex optimization problem (attack objective) and gradient-based optimizers can easily be stuck at local minimums. We have also demonstrated in the first two paragraphs of section 3 that white-box attacks are also sensitive to the initial point.
>
> In fact, it has been shown in (Katz et al., “Reluplex: An Efficient SMT Solver for Verifying Deep Neural Networks”) that for a ReLU network, finding the optimal attack is NP-hard (see their Appendix I), and an exponential time algorithm proposed in the same paper can only scale to networks with <200 neurons. So it’s computationally impossible to find the optimal attack for the MNIST/CIFAR/ImageNet networks used in our paper.
>
> Moreover, it’s NP-complete to find the optimal attack for the tree-based model (see Kantchelian et al., “Evasion and Hardening of Tree Ensemble Classifiers”, Section 4.2).

---

> ### Author Response · Authors · 2019-11-15
> **Response to Reviewer 1  # Part 2**
>
> [Time complexity]
>
> The time complexity ( number of queries regarding the parameters mentioned in Algorithm 1 is relatively simple. We briefly discuss it in Appendix H.
>
> [Figure 1]
> Sorry for the confusion. In the description of Figure 1, a local minimum indicates a point on the decision boundary that has the shortest distance to the original example, compared with other nearby points on the decision boundary. Those local minimums are the points where a decision-based attack can converge to.
>
> In Figure 1, we plot the decision boundary for a given model around the original example. The decision boundary is a high-dimensional surface, so we plot its projection to a 2D tangent plane. To choose which 2D hyperplane to project to, we run a decision-based attack from two random initialization points, and use their converged perturbation directions as the vector to form the 2D hyperplane. These two directions are presented as red lines in Figure 1, showing that they are pointing to local minimums on the surface (local minimum in terms of distance to the original example). We chose the hyperplane by this way since it guarantees that there are at least 2 local minimums on this hyperplane.  We have revised the paper to make this more clear.
>
> [minimize l(.) / g(.) in Algorithm 2]
> There is a mistake in Algorithm 2, we should minimize $g(.) / l(.)$, sorry for the confusion! We have corrected this in the revision.
>
> [Theorem 1]
> Theorem 1 in Appendix B.2 is the results of Bergstra et al.'s. We put it there so the readers can understand that more easily.
>
> [grammar errors]
> Thank you for the editorial comments we correct them in the revised version.

---

### Decision · Program_Chairs · 2019-12-19

**Decision:**

Reject

**Comment:**

This paper proposes BOSH-attack, a meta-algorithm for decision-based attack, where a model that can be accessed only via label queries for a given input is attacked by a minimal perturbation to the input that changes the predicted label. BOSH improves over existing local update algorithms by leveraging Bayesian Optimization (BO) and Successive Halving (SH). It has valuable contributions. But various improvements as detailed in the review comments can be made to further strength the manuscript.